# Impact of Bis-*O*-dihydroferuloyl-1,4-butanediol Content on the Chemical, Enzymatic and Fungal Degradation Processes of Poly(3-hydroxybutyrate)

**DOI:** 10.3390/polym14081564

**Published:** 2022-04-11

**Authors:** Quentin Carboué, Sami Fadlallah, Yasmine Werghi, Lionel Longé, Antoine Gallos, Florent Allais, Michel Lopez

**Affiliations:** URD Agro-Biotechnologies Industrielles (ABI), CEBB, AgroParisTech, 3 Rue des Rouges-Terres, 51110 Pomacle, France; yasmine.werghi1@gmail.com (Y.W.); lionel.longe@agroparistech.fr (L.L.); antoine.gallos@agroparistech.fr (A.G.); florent.allais@agroparistech.fr (F.A.); michel.lopez@agroparistech.fr (M.L.)

**Keywords:** poly-β-hydroxybutyrate, plasticizer, biodegradation, imagery analysis, *Actinomucor elegans*

## Abstract

Poly-β-hydroxybutyrate (PHB) is a very common bio-based and biocompatible polymer obtained from the fermentation of soil bacteria. Due to its important crystallinity, PHB is extremely brittle in nature, which results in poor mechanical properties with low extension at the break. To overcome these issues, the crystallinity of PHB can be reduced by blending with plasticizers such as ferulic acid derivatives, e.g., bis-*O*-dihydroferuloyl-1,4-butanediol (BDF). The degradation potential of polymer blends of PHB containing various percentages (0, 5, 10, 20, and 40 w%) of BDF was investigated through chemical, enzymatic and fungal pathways. Chemical degradation revealed that, in 0.25 M NaOH solution, the presence of BDF in the blend was necessary to carry out the degradation, which increased as the BDF percentage increased. Whereas no enzymatic degradation could be achieved in the tested conditions. Fungal degradation was achieved with a strain isolated from the soil and monitored through imagery processing. Similar to the chemical degradation, higher BDF content resulted in higher degradation by the fungus.

## 1. Introduction

The global environmental impact and the growing social concerns regarding conventional plastics, accompanied by the high rate of depletion of petroleum resources and the evolution of legislation, have led to the development of bio-based polymers [1]. However, the mere nature and renewable origin of bio-based polymers obtained from biomass are not sufficient to qualify a material as “green”. Indeed, many other criteria have to be considered, especially regarding its synthesis conditions and (bio)degradation [2]. Further, bio-based polymers are mostly used in packaging applications because of their higher costs of production and their significantly lower performances in comparison with petroleum-based materials [3]. In addition, confusion may exist regarding the bio-based and biodegradable aspects: not all bio-based polymers are biodegradable as some synthetic bio-based polymers exhibit very low biodegradation potentials. Some critical factors involved in the degradation of polymers are molecular weight, crystallinity, wettability, porosity, and material thickness [4,5,6,7,8]. On the other hand, all natural polymers are biodegradable [9].

Polyhydroxyalkanoates (PHA), such as poly(3-hydroxybutyrate) (PHB), are intracellular carbon and energy storage compounds (carbonosomes) found in some soil bacteria such as *Cupriavidus necator* [10]. It is a thermoplastic with similar properties to petroleum-based polypropylene with a melting point (*T_m_*) of 180 °C [11]. It can be produced through fermentation using renewable and low-cost biomass feedstocks such as wastewater from the sugar industry, permeate from the dairy industry, and lignocellulosic byproducts as culture media [12,13]. Further, PHB is a biocompatible polymer whose biocompatibility can be increased through rapid lipase- or sodium hydroxide-based treatments [14]. However, PHB, because of its high crystallinity, is a brittle material due to a low glass transition temperature (*T_g_*) (i.e., *T_g_* of pure PHB is between 2 and 15 °C) and a secondary crystallization of the amorphous phase that occurs at room temperature [15,16]. The focus has been made to find compounds that improve the mechanical and chemical properties of PHB (e.g., notably improving the flexural strength, impact strength, heat deflection, long-termed stability) while maintaining its biodegradability and biocompatibility levels. PHB copolymers involving 4-hydroxybutyrate, agave fibers, or lignin, have been developed and exhibited higher mechanical properties than neat PHB [17,18,19]. Another more economical way to improve the mechanical properties of PHB is through the formation of blends using plasticizers such as the PHB-glycerol blend, which shows a reduction in crystallinity when compared with PHB homopolymer [15]. It has been previously demonstrated that bio-based bis-*O*-dihydroferuloyl-1,4-butanediol (BDF)—obtained from the highly selective lipase-catalyzed enzymatic reaction of two ferulic acids with 1,4-butanediol—blended with PHA significantly improved the mechanical properties when compared with the polymer alone [20,21].

The current article aims to study the degradation processes and patterns of a blend of PHB with various incorporation levels of BDF (5, 10, 20, 30, and 40 w%) through various modalities: chemical, enzymatic and fungal to study the impact of BDF contents on the (bio)degradation processes. Indeed, as new polymeric compounds are produced, it is important to evaluate their (bio)degradation following different strategies and conditions to understand the underlying mechanisms and evaluate the most effective management strategy regarding environmental, economic, and societal considerations. In the case of fungal biodegradation, the monitoring of the evolution of the polymeric particles was carried out through image analysis of the binary images obtained after the treatment of pictures. This is an original approach as image analysis is usually used to evaluate the structural properties of various polymeric materials [22,23,24].

## 2. Materials and Methods

### 2.1. Polymers

Plasticizer-free PHB powder (Batch T19) was supplied by BIOMER^©^ (Schwalbach am Taunus, Germany). Ferulic acid was acquired from Biosynth-Carbosynth^©^ (Thal, Switzerland). BDF was synthesized following the protocol of Pion et al. (2013) [20]. The structures are presented in Figure 1.

#### 2.1.1. Preparation of PHB/BDF Specimens

Extrusion of PHB/BDF blends was performed on a compounding extruder HAAKE MiniLab II (Thermo Fisher Scientific^©^, Waltham, MA, USA) twin screw, screw diameter 16 mm, 24 mm. Screws were set in co-rotation, at 60 rpm at an extrusion temperature of 170 °C. HAAKE MiniJet Pro Piston Injection Molding System was used for the injection molding of samples specimen. A DMA test bar mold was used, with dimensions of 60 mm × 10 mm × 1 mm. The mold was maintained at 45 °C during the injection.

#### 2.1.2. Preparation of PHB/BDF Films

The PHB films were formed using solvent casting: specimens of PHB/BDF blends were dissolved in chloroform (VWR Chemicals BDH^©^, Radnor, PA, USA) and the solution was then poured into stainless steel rectangular molds (11 × 6 cm). The solvent was then slowly evaporated under the fume hood at room temperature to obtain uniform films of around 0.1 µm thickness as measured by an electronic digital caliper.

#### 2.1.3. Preparation of PHB/BDF Powders

Around 5 g of the specimen of PHB/BDF blend were ground using stainless steel marble agitated at a frequency of 20 Hz at −196 °C for 2 min using a Cryomill (Retsch GmbH^©^, Haan, Germany).

#### 2.1.4. Characterization of the Material

##### Differential Scanning Calorimetry

Differential Scanning Calorimetry (DSC) was performed with a DSC Q20 (TA Instruments^©^, New Castle, DE, USA). Around 8 mg sample was placed in a sealed pan and flushed with highly pure nitrogen gas. The first heating ran from −80 °C to 200 °C with a heating rate of 10 °C·min^−1^ was performed to determine the melting temperature (*T_m_*) and the fusion enthalpy (ΔH_f_). This is followed by a cooling run to −80 °C with a cooling ramp of 200 °C·min^−1^. The glass transition temperature (*T_g_*) was obtained in the second heating run from −80 °C to 200 °C at 10 °C·min^−1^. Indeed, PHB tends to crystallize during the cooling step. The rate of the cooling was thus increased to quench the material and avoid its recrystallization easing this way the measurement of the *T_g_*.

##### Fourier-Transform Infrared Spectroscopy

Fourier-transform infrared (FT-IR) spectra were recorded from film samples in the middle infrared (4000–650 cm^−1^) using a Cary 630 FTIR Spectrometer (Agilent Technologies^©^, Santa Clara, CA, USA) working in passing mode.

### 2.2. Enzymatic and Chemical Degradation

#### 2.2.1. Chemical Degradation

Specimens of PHB/BDF blends (with BDF content of 0, 5, 10, 20, 30, and 40 w%) weighting 100 mg were put in a 4 mL of 0.25 M NaOH solution. The mixture was stirred at 100 rpm at room temperature for 32 h. HCl (3 M) was then added until reaching an acidic pH to stop the reaction. The obtained precipitate was washed with distilled water and the polymer was then dry-frozen prior to analysis. In order to confirm the results, the experiment was also carried out with commercial non-extruded PHB (Sigma-Aldrich^©^, Saint-Louis, MO, USA) powder. A reaction kinetic was also carried out for the PHB/BDF blend with (40 w% of BDF) with a sampling every 30 min during the first 9.5 h of alkali degradation reaction.

#### 2.2.2. Enzymatic Degradation

The evaluation of the enzymatic biodegradation potential of the various PHB/BDF blends was carried out following the protocol used by Rodríguez-Contreras et al. (2012) [25]. Briefly, a suspension of 17 mg·mL^−1^ of powder blend (with BDF content of 0, 5, 10, 20, 30, and 40 w%) was put to react with 17.5 mg of Lipopan^©^ 50 BG (1,3 specific lipase from *Thermomyces lanuginosus*; Novozymes^©^, Bagsværd, Denmark) into hemolytic tubes containing 3 mL of phosphate buffer (0.05 M, pH = 7.4) at 37 °C, agitated at 50 rpm for a week. The same protocol was repeated for films of PHB blends and with commercial PHB (Sigma-Aldrich^©^) powder. Some additional reaction parameters were also investigated, a pH of 9, a temperature of 45 °C, an absence of stirring, and a two-fold and ten-fold concentration of enzyme. Another industrial lipase was also tested: Eversa^©^ (free liquid lipase from genetically modified *Aspergillus oryzae*; Novozymes^©^), following the previous protocol but at a temperature of 40 °C. The tube was then put at 0 °C to stop the reaction, and centrifuged at 4750 rpm for 10 min. The obtained precipitate was washed with distilled water, resuspended, and centrifuged again to remove the enzyme. The polymer was then lyophilized prior to analysis. Enzyme activities were checked using the p-nitrophenyl butyrate colorimetric assay. Absorbance was monitored at 400 nm and the results were compared with a p-nitrophenol calibration curve.

#### 2.2.3. Evaluation of the Degradation Potential through Nuclear Magnetic Resonance (NMR) Spectroscopy

Degradation of both chemical and enzymatic ways was investigated using NMR. ^1^H NMR spectra were recorded on a Bruker Fourier 300 MHz (CDCl_3_ residual signal at 7.26 ppm). Data are reported in Appendix A.

All NMR assignments were also carried out using ^1^H-^1^H COSY (Appendix A). The following peaks were followed: 4.20 ppm corresponding to β-hydroxybutyric acid (the constituting monomer of PHB) and 5.20 ppm corresponding to the PHB chain. Polymer degradation at a certain time t is then expressed as the ratio of the appearance of the integrated monomer peak to the disappearance of the integrated polymer peak.
(1)$$\mathrm{Degree}\;\mathrm{of}\;\mathrm{degradation}\left(t\right)\left(\%\right)=\frac{{\left[\mathrm{monomer}\;\mathrm{peak}\right]}_{t}}{{\left[\mathrm{monomer}\;\mathrm{peak}\right]}_{t}{+[\mathrm{polymer}\;\mathrm{peak}]}_{t}}\times 100$$

With ${\left[\mathrm{monomer}\;\mathrm{peak}\right]}_{t}$ and ${\left[\mathrm{polymer}\;\mathrm{peak}\right]}_{t}$ corresponding to the integrated ^1^H NMR of the monomer and polymer peaks at time t, respectively.

### 2.3. Fungal Degradation

#### 2.3.1. Sample Collection

Two soil samples were collected in sterilized 1 L plastic bottles from two sites. One on the banks of the brook Le Petit Ru (Bazancourt, France) and one from the embankment of the highway (E420, Pomacle, France). Bigger fractions involving plastic pieces, wood, rocks, and leaves were removed and the soil samples were then kept at room temperature in the dark.

#### 2.3.2. Isolation of PHB-Degrading Fungi

Five films were buried vertically at a depth of 5 cm in each soil sample complemented with 200 mL of a solution containing 2 g of glucose and incubated in a lab oven at 30 °C. After 4 weeks, films were recovered from the soil and placed on Petri dishes containing Sabouraud (SAB) agar medium (Sigma-Aldrich^©^). The Petri dishes were incubated at 30 °C for 4 days. The different fungal strains were then isolated on different SAB Petri dishes. To evaluate the PHB degradation potential of the various isolated fungi, each strain was cultivated on a medium containing PHB as the sole carbon source. The composition of the selective medium was: 11.28 g·L^−1^ of M9 Minimal Salts 5× medium (Sigma-Aldrich^©^), 0.011 g·L^−1^ of calcium chloride, 0.24 g·L^−1^ of magnesium sulfate, 0.02 g·L^−1^ of yeast extract and 15 g·L^−1^ of agar and 1 g·L^−1^ of PHB powder. One strain was able to grow on the selective medium with PHB as a sole carbon source and was identified as *Actinomucor elegans* at the Muséum National d’Histoire Naturelle de Paris (France). The purity of a 3-days’ culture was tested with a DNEasy Plant mini Kit (Qiagen^©^, Venlo, The Netherlands). Internal transcribed spacer region amplification was carried out through polymerase chain reaction using ITS5 and ITS4 primers. After sequencing, taxonomic affiliation was made using BLAST and GenBank (National Center for Biotechnology Information, Bethesda, MD, USA).

#### 2.3.3. Evaluation of the Degradation Potential of PHB

The strain of *A. elegans* was grown on a SAB plate containing a PHB film of 2 cm in length. After 3 weeks, the mycelium was gently removed using a scalpel. The film was rinsed with distilled water and dried at 60 °C for 24 h before being analyzed through FT-IR and DSC.

#### 2.3.4. Evaluation of the Degradation Potentials of the PHB Blends through Image Analysis

The PHB degradation ability of the isolated fungal strains able to grow on the PHB-containing selective medium was then investigated through image processing and analysis. Selective medium containing PHB powder with various percentages of BDF (0, 5, 10, 20, and 40 w%) were inoculated with 4 strains using 6-well plates. For 14 days, photos were taken using an Ebox CX5.TS (Vilber Lourmat^©^, Marne-la-Vallée, France). For the same kinetic series, the obtained pictures were then converted into binary images with the same threshold value in order to only visualize the PHB/BDF grains as white pixels. The evolution of the blend grains was then analyzed by observing the total white pixels in the matrixes of the binary pictures during the time. The pictures were treated using Illustrator (Adobe^©^, San José, CA, USA) and the image processing was carried out using the thresholding application in MATLAB (MathWorks^©^, Natick, MA, USA) [26]. Polymer degradation at a certain time t is then expressed as the percentage of the initial total white pixels (WP) of the binary matrix.
(2)$$\mathrm{Degree}\;\mathrm{of}\;\mathrm{degradation}\left(t\right)\left(\%\right)=\frac{{\mathrm{WP}}_{t}}{{\mathrm{WP}}_{t_{0}}}\;\times \;100$$

With ${\mathrm{WP}}_{t}$ the white pixels of the binary image at t and ${\mathrm{WP}}_{t_{0}}$ the white pixels of the photo at $t_{0}$.

To compare the degradation kinetics of the various PHB/BDF blends, the results for each sample were also expressed as t_50_, which is the duration necessary to reach 50% of the degradation of the initial polymer. The t_50_ values were determined using polynomial regression with the highest significance (the selected polynomial degree for the model was significant when tested with an F-test and maximized both the coefficient of determination R² and the adjusted coefficient of determination adj-R^2^).

#### 2.3.5. Statistical Analysis

Experiments were carried out in triplicate. The results were tested by one-way ANOVA and the comparison of the means was done using Tukey post hoc range tests (α= 0.05). Statistical analyses were carried out using MATLAB (MathWorks^©^).

## 3. Results and Discussion

Along with the article, the polymer materials are tested under different forms—specimen, film, and powder—because certain forms are more suitable than others to carry out the tests. The thermal properties of the tested materials under the different forms are given in Table 1.

### 3.1. Chemical Degradation

The study of the degradation of the polymer blends through the use of a chemical agent was carried out to serve as a model to study the degradation processes with the other methods. Sodium hydroxide was chosen because it is a strong stress cracking agent to PHB. Indeed, it has been shown that a solution of 3 M NaOH dramatically damaged the surface and favored the surface cracking of PHB-HV (96/6%) during tensile testing [27]. However, at lower concentrations of 0.1 to 1 M, alkali treatments of 24 h increase the hydrophilicity of PHB film surface through breaking of ester bonds and subsequent increase in densities of hydrophilic terminal carboxyl and hydroxyl groups without changing the overall mechanical properties as suggested by the weight loss values that were 0.3 and 3.7 w% for the 0.1 to 1 M treatments, respectively [28]. Following this idea, Yang et al. (2002) have shown that PHB films treated with 1 N NaOH solution at 60 °C for 1 h reduced the pore sizes of the films, although the authors did not observe any macroscopic morphological change on the surfaces. This modification of the texture of the film, coupled with the increase in the hydrophilicity improved the biocompatibility of the films as the mammal cells were more likely to attach on the surface to grow [14,29]. In the present case, alkali treatments (0.25 M NaOH solution) were carried out on the specimen. After 32 h, degradation was only observed for the samples that contained BDF. Furthermore, the results showed that as BDF increases, the percentage of degradation increases as well—23.0% and 73.7% for the PHB/BDF blends 5 w% and 40 w%, respectively—suggesting an important role of the plasticizer in the hydrolysis reaction (Figure 2).

This aspect may be due to the altered properties of the blends in the presence of BDF. This renders the polymer chains of PHB more susceptible to NaOH hydrolytic action (Table 1). A possible mechanism is related to the saponification of BDF itself that liberates oxyanions and carboxylates that may participate in the PHB chains scissions in a bulk degradation process (Figure 3). On the contrary, no degradation was observed for the sample with 100 w% PHB. This result was confirmed by the fact that, in the case of the commercial PHB powder, no alkaline degradation could be observed either, suggesting that extrusion had no influence on the degradation process. Interpretations of the spectra were also made by comparison with the monomer (β-hydroxybutyric acid) spectrum (Appendix A). Confirmation of the presence of monomers in the spectra was carried out with homonuclear correlation spectroscopy (^1^H-^1^H COSY) (Appendix A). The kinetics of the spectra during the alkali degradation showed a progressive appearance of the monomer—peak at 4.2 ppm—and disappearance of the polymer—peak at 5.2 ppm—(Figure 4).

### 3.2. Enzymatic Degradation

The specific enzymes involved in PHB degradation are the PHB-depolymerase. There are two types of such enzymes: intracellular PHB-depolymerases for which the PHB represents the endogenous carbon reservoir and the hydrolysis is carried out by an accumulation of bacteria, and extracellular PHB-depolymerases for which PHB is an exogenous carbon source and is not limited to the producing bacteria but is also largely found in fungi [16]. In comparison, the occurrence of extracellular depolymerases is more widely reported amongst microorganisms as compared to that of intracellular depolymerases [30]. However, the number of commercially available PHB-depolymerases is still scarce as are the reports of commercial hydrolases that efficiently degrade PHB. Indeed, commercial lipases, esterases, and proteinases from various sources have shown no degradation effects on PHB [31]. In the present case, and unlike the study carried out by Alejandra et al. (2012), no degradation has been observed on any PHB film regardless of the BDF content during our enzymatic reactions carried out with Lipopan^©^ 50 BG and Everva^©^ [25] (Appendix A). This absence of enzymatic degradation was observed regardless of the nature of the blends—powder or film. The extrusion also had no effect as the commercial PHB was not degraded either. Moreover, the tested pH (7.4 and 9), the tested temperatures of reaction (37 °C and 45 °C), the stirring (50 rpm) or the static conditions and the tested enzyme concentrations (17.5, 35, and 175 mg·mL^−1^) had no effect on the process.

### 3.3. Fungal Degradation

To carry out an isolation of PHB-degrading microorganisms, the film was chosen over the specimen because it maximizes the surface available for the microorganisms for colonization and, unlike powder, it can be recovered for further isolation. Soil samples were chosen as the source of inoculum because it is the natural environment of PHB-producing bacteria. This way, a study involving the isolation of PHB-degrading microorganisms from various habitats such as compost, soil, air, and horse dung has shown a prevalence of soil fungi over bacteria, protozoa, and lichens as potential degraders [32]. This is not surprising since fungi are primary decomposers of polymeric compounds in many ecosystems. Identified PHB-degrading fungi involve fungi of the genera *Paecilomyces*, *Aspergillus*, *Penicillium*, *Fusarium*, *Alternaria,* and *Trichoderma* [32,33,34,35]. After 4 weeks in the soils, the samples of PHB films were highly degraded, fragmented on the edges, and with variable thickness. The cultivation of these fragments of PHB film led to the isolation of 45 different strains. From this collection, further isolation using a selective medium containing PHB as the sole carbon source allowed us to isolate a strain of zygomycete belonging to the order of the Mucorales: a strain of *A. elegans* (Appendix A). This species has previously been reported as a member of the fungal community colonizing PLA/PHB blend mulches in the soil [36]. The ability of the fungus was confirmed through FT-IR analysis of a PHB film after the *A. elegans* was grown on it for 3 weeks (Figure 5). After the cultivation of the fungus, there is the appearance of a large band between 3000 and 3600 cm^−1^, corresponding to the hydroxide and carboxylic acid functions carried by the oligomers after the cleavage of PHB chains. Moreover, bands located at 1634 cm^−1^ also appeared after degradation, suggesting the formation of carbon–carbon double bonds in the backbone of the molecules, and the band at 1716 cm^−1^ corresponding to the carbonyl ester group was reduced greatly. The conversion of carboxylic acid groups into alkenes has already been highlighted during fungal degradation [37].

After having selected the strain and confirmed its potential in the degradation of PHB, the effect of BDF composition in the blends was investigated using imagery analysis of the pictures that were taken during the fungal growth. Indeed, in the media containing the different PHB/BDF blends as the sole source of carbon, the PHB/BDF particles are visible as white grains and this form is thus convenient to measure their evolutions during the time, as the fungus grows (Figure 6).

In order to quantify the polymer degradation, the pictures were treated. Binarization is the conversion process of a multi-tone image into a bi-tonal image in which a threshold range is set; associating pixels to white (or 1) if its tone value is within the threshold boundaries, or to black (0) if it is outside the range. The generated binary matrixes ease the mathematical treatments, notably decreasing the computational load [38] (Figure 7). The selection of the threshold range is a delicate step as many factors may affect the final binary image beforehand, that is why it is important to standardize the shot of films with the same illumination, exposure time, and focal distance. Moreover, the mycelium being white may also interfere with the final calculation of the WP in the binary image. For that matter, the thresholding function is useful as it allows the user to set the threshold interval in order to find the right compromise in the grey tones acceptation to only consider PHB/BDF particles and reject mycelium from the binary image. In the present case, the image treatment had to involve the same threshold range on the complete photo series constituted by a degradation kinetic to obtain relative values regarding this degradation.

The results obtained from the image treatment and analysis revealed an influence of the BDF percentage on the degradation of the blend by the fungus (Figure 8). This was statistically verified by comparing the t_50_ that decreases as the BDF percentage increases (Figure 9). Pure PHB was slower to be degraded by *A. elegans* as it required more than 300 h to be fully degraded. On the other hand, the presence of more than 20 w% of BDF in the blend showed more rapid degradation of the material by the fungus as 50% of degradation was reached at around 100 h. It is worth mentioning that many fungi with plant-degrading activities are able to degrade and metabolize p-hydroxycinnamic acid derivatives such as ferulic acid, which is a structural compound in the lignocellulosic matrix [39,40]. In the present case, it seems that *A. elegans* preferentially degraded the PHB blends with higher BDF content. By contrast, blends of lignin and PHB obtained by melt extrusion were more resistant to microbial degradation after being buried in the soil. This phenomenon has been explained by the fact that lignin formed strong hydrogen bonds with the PHB, thus reducing its bioavailability for microbes [41].

## 4. Conclusions

Whatever the tested conditions, no enzymatic biodegradation of the PHB/BDF blends has been achieved in this work using industrial lipases. The chemical and fungal (bio)degradation processes of the PHB blends, on the other hand, were enhanced as higher contents of the plasticizer BDF were used. In the chemical degradation experiments in NaOH solution (0.25 M), it was shown that BDF was essential for the degradation to occur. In the case of fungal biodegradation, a new strain was isolated from soil samples and the evaluation of the fungus-mediated biodegradation was carried out through an image processing approach, which allows an indirect, non-destructive online monitoring of the biodegradation. Nevertheless, other monitoring techniques can be used to evaluate the metabolic activity during fungal growth, such as respirometric measurements of the consumed O_2_ or the produced CO_2_. This approach holds an interesting perspective since the experiments were carried out in well plates with the potential development of automated procedures, to use this approach in high-throughput screening studies. Regarding the *A. elegans* strain, evaluation of its metabolic potential could eventually result in the production of interesting high-value molecules from PHB residues and further evaluations should include proteomic studies to determine which enzymes are involved in the PHB degradation.

## Figures and Tables

**Figure 1 polymers-14-01564-f001:**
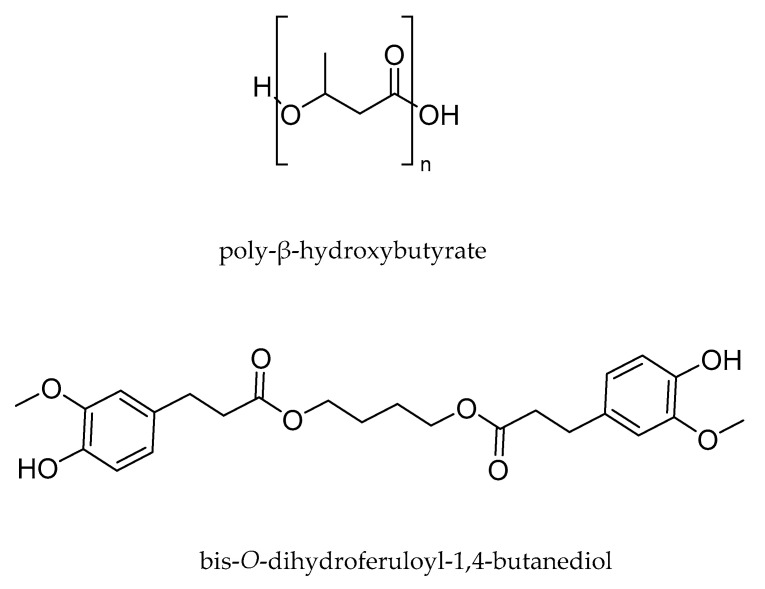
Structures of PHB and BDF.

**Figure 2 polymers-14-01564-f002:**
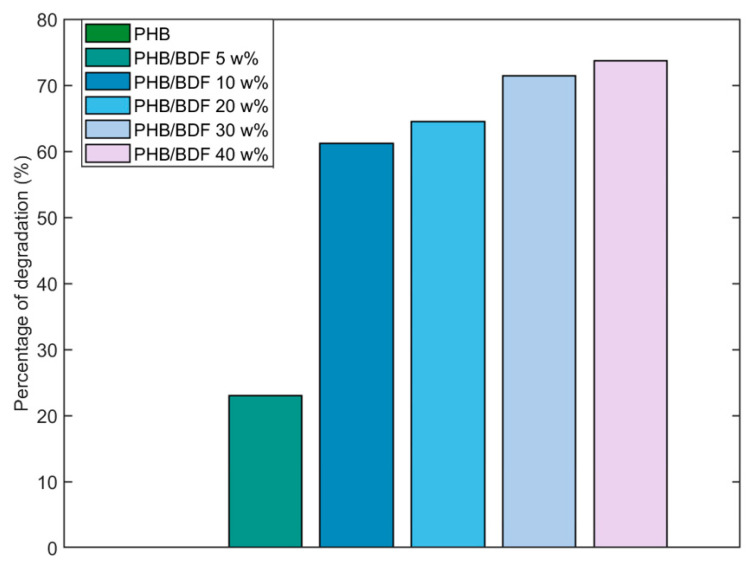
Degradation of specimens of PHB/BDF blends in a NaOH solution (0.25 M) during 32 h.

**Figure 3 polymers-14-01564-f003:**
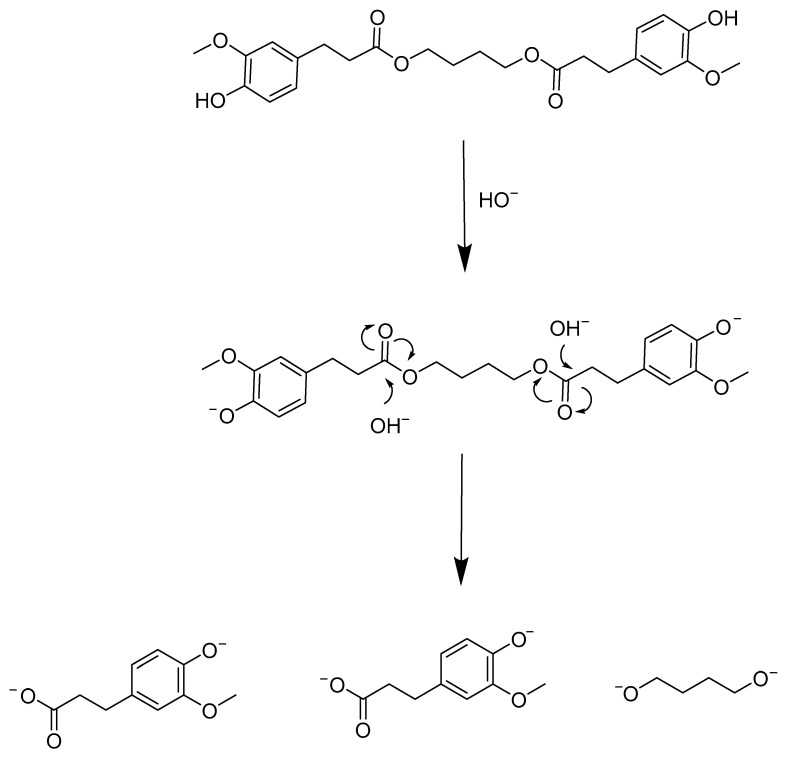
Saponification reaction with BDF.

**Figure 4 polymers-14-01564-f004:**
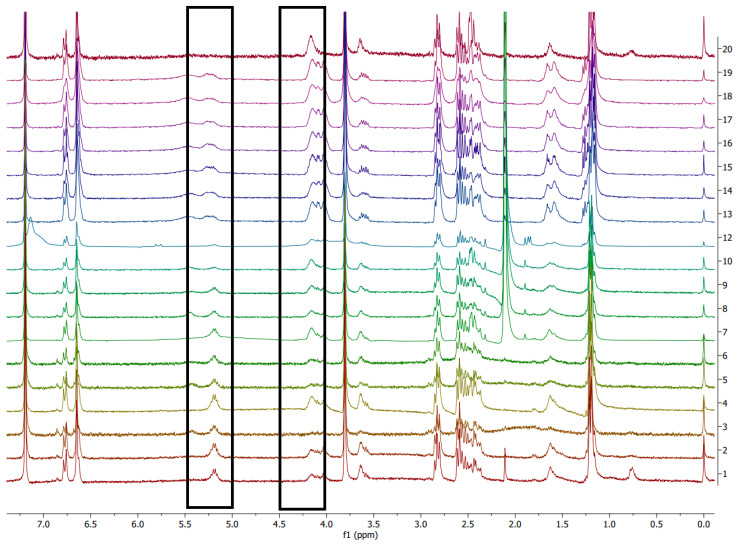
^1^H NMR (CDCl_3_) spectrum of specimen PHB/BDF 40 w% during alkali degradation with NaOH solution (0.25 M), spectrum 1 corresponds to t_0_; up to spectrum 19, samples were taken every 30 min of reaction; spectrum 20 corresponds to 32 h reaction.

**Figure 5 polymers-14-01564-f005:**
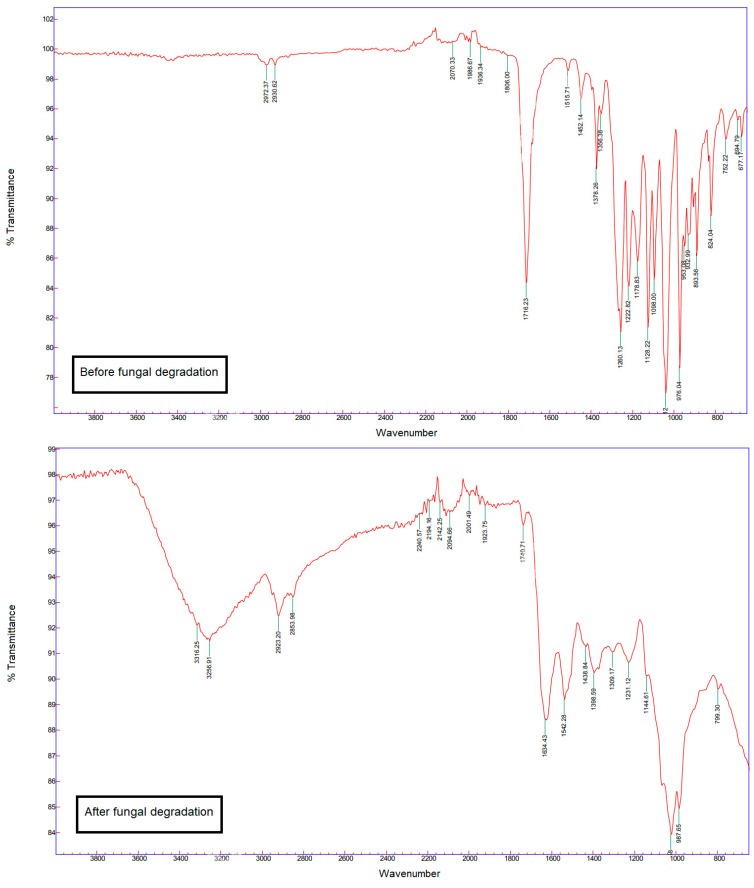
FT-IR spectra of the film before and after 21 days of colonization by the *A. elegans* strain.

**Figure 6 polymers-14-01564-f006:**
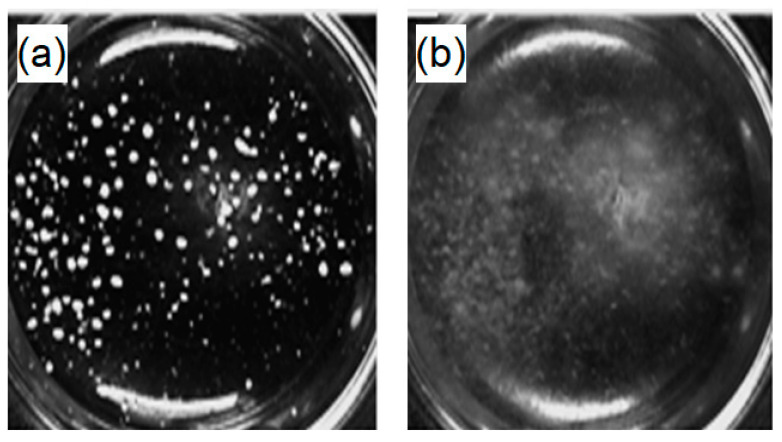
Pictures of a well containing the selective medium with powder of PHB/BDF blend as a sole carbon source: (**a**) medium at t_0_ and (**b**) after 240 h of fungal growth.

**Figure 7 polymers-14-01564-f007:**
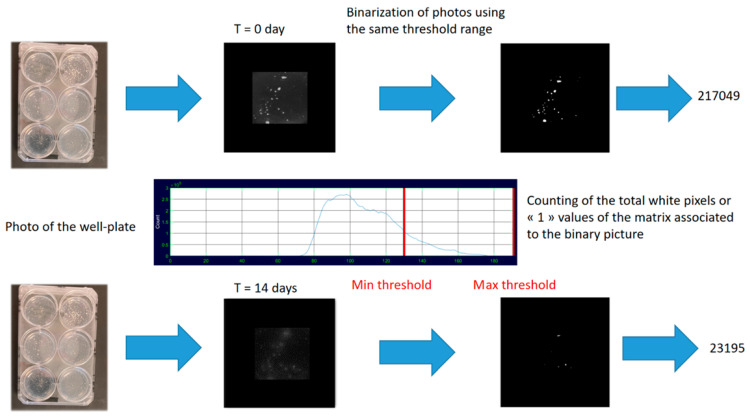
Image treatment and analysis process.

**Figure 8 polymers-14-01564-f008:**
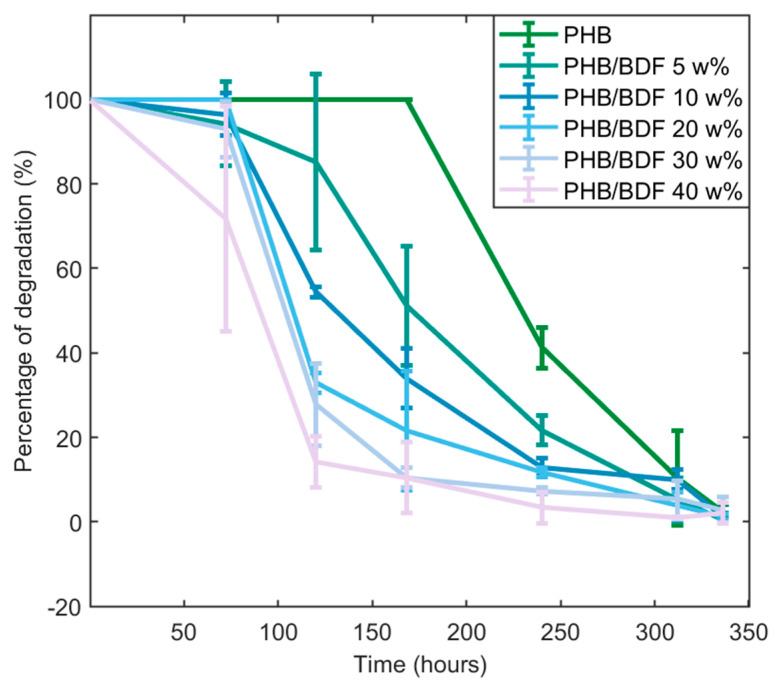
Degradation kinetics of each powder of PHB/BDF blends obtained from image analysis.

**Figure 9 polymers-14-01564-f009:**
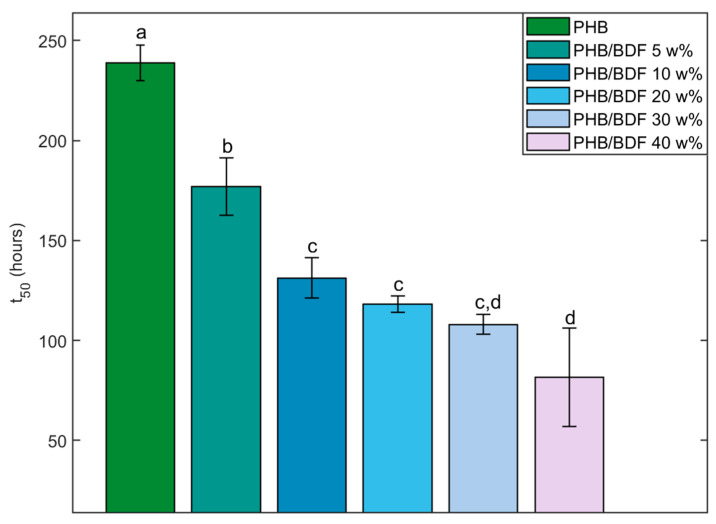
t_50_ associated to each powder of PHB/BDF blend. Different letters mean a significant difference as determined with a one-way ANOVA and Tukey post hoc range tests (α = 0.05).

**Table 1 polymers-14-01564-t001:** DSC Analysis of the PHB/BDF Blends.

Type	Specimen				Film				Powder			
Composition	*T_g_* (°C)	*T_m_* (°C)	*T_c_* (°C)	*T_cc_* (°C)	*T_g_* (°C)	*T_m_* (°C)	*T_c_* (°C)	*T_cc_* (°C)	*T_g_* (°C)	*T_m_* (°C)	*T_c_* (°C)	*T_cc_* (°C)
PHB	0.7	162.9	90.4	35.8	0.1	159.8	-	36.5	-	170.0	92.2	-
PHB/BDF 5 w%	0.0	162.0	64.1	36.2	−4.7	157.7	-	32.5	0.75	160.7	66.6	31.7
PHB/BDF 10 w%	−1.5	156.7	-	37.2	0.8	160.1	-	39.5	0.7	158.5	58.7	34.5
PHB/BDF 20 w%	−5.8	151.4	-	38.0	−2.5	152.7	-	39.8	−4.9	150.6	-	35.4
PHB/BDF 30 w%	−8.6	146.7	-	44.3	−11.7	140.0	-	42.4	−8.4	143.3	-	35.9
PHB/BDF 40 w%	−9.4	144.0	-	45.7	−10.3	145.5	-	37.4	−9.1	142.2	-	38.0

## Data Availability

Data sharing not applicable.

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
