# Peer review of "Impact of Bis-O-dihydroferuloyl-1,4-butanediol Content on the Chemical, Enzymatic and Fungal Degradation Processes of Poly(3-hydroxybutyrate)"

_polymers, 2022, doi:10.3390/polym14081564_

Round 1

Reviewer 1 Report

The research presented by the Authors is quite interesting and covers issues related to the degradation of PHB, and more precisely the influence of the plasticizer on this process. The article may be published, however, minor changes/additions are needed before final approval.

  1. Line 102-108. Please explain why the parameters were determined from different heating curves.
  2. Line 110 – 111. Could more information be provided on FTIR studies? In what form were the samples tested. In what mode - reflected, passing?
  3. 2, Fig. 8, Fig. 9 - I would consider changing the colors of individual samples to more contrasting ones. On some of my devices, the colors were too close and difficult to distinguish.
  4. Line 298 – 301. The spectra before and after degradation differ significantly. The authors describe only two peaks. Is the appearance/disappearance of other bands not also significant?

Author Response

Answers to reviewers’ report (Manuscript ID: polymers-1661901)

Daer reviewer,

First of all, my co-authors and myself would like to thank you for having allowed us to submit a revised manuscript.

We are also very grateful to you for having reviewed our manuscript. All the modifications have been highlighted using the track change function.

Please find below the answers to your comments.

Reviewer 1:

Comments and Suggestions for Authors

The research presented by the Authors is quite interesting and covers issues related to the degradation of PHB, and more precisely the influence of the plasticizer on this process. The article may be published, however, minor changes/additions are needed before final approval.

Thank you for your comments.

Line 102-108. Please explain why the parameters were determined from different heating curves.

This part indeed was lacking explanations, we added the following to clarify the use of different cycles: “Indeed, PHB tends to crystallize during the cooling step. The rate of the cooling was thus increased to quench the material and avoid its recrystallization easing this way the measurement of the Tg.”.

Line 110 – 111. Could more information be provided on FTIR studies? In what form were the samples tested. In what mode - reflected, passing?

We added precisions concerning the type of sample that was analyzed through FT-IR and the mode of analysis.

2, Fig. 8, Fig. 9 - I would consider changing the colors of individual samples to more contrasting ones. On some of my devices, the colors were too close and difficult to distinguish.

You are right, we modified the colors accordingly creating a gradient based on green, blue and pink.

Line 298 – 301. The spectra before and after degradation differ significantly. The authors describe only two peaks. Is the appearance/disappearance of other bands not also significant?

Indeed, we decribed better the comparison between the two conditions in the context of fungal degradation.

Reviewer 2 Report

The article was devoted to a very actual topic of the biosynthesis of renewable biocompatible polymers that can be used for the manufacture of packaging materials with a guaranteed probability of their decomposition in the soil. Authors used poly-β-hydroxybutyrate (PHB) with the addition of a plasticizer bis-O-dihydroferuloyl-1,4-butanediol (BDF). The ability to decompose PHB/BDF blends under the action of chemical, industrial lipases and fungal cultures at various concentrations of the plasticizer was studied. Particular advances have been made in assessing biodegradation in the case of fungal degradation using image processing, as well as monitoring of O2 consumption and CO2 production. Although no biodegradation was found with industrial lipases, it was found with chemical and fungal degradations, and this effect increased with increasing of plasticizer. Without any doubt, the article will be of interest for the readers of Polymers.

I think that the article can be improved taking into account the following comments and suggestions:

- The strength of the study is that the authors used three types of blends: specimen, film and powder. It is necessary to clearly indicate in the captions to the figures in which cases which types of samples were used (Figs. 2,4,5,6,8,9).

- Lines 150, 159; Equations 1, 2. Why not use the term Degree of Degradation instead of Degradation.

- What can be the accuracy of determining the values presented in Fig.2. Is the difference between samples with 30 and 40% plasticizer statistically significant?

- It is better to reduce the size of the drawings. It is necessary to correct expressions like g.L-1 to  g∙L-1.

Author Response

Answers to reviewers’ report (Manuscript ID: polymers-1661901)

Daer reviewer,

First of all, my co-authors and myself would like to thank you for having allowed us to submit a revised manuscript.

We are also very grateful to you for having reviewed our manuscript. All the modifications have been highlighted using the track change function.

Please find below the answers to your comments.

Reviewer 2:

Comments and Suggestions for Authors

The article was devoted to a very actual topic of the biosynthesis of renewable biocompatible polymers that can be used for the manufacture of packaging materials with a guaranteed probability of their decomposition in the soil. Authors used poly-β-hydroxybutyrate (PHB) with the addition of a plasticizer bis-O-dihydroferuloyl-1,4-butanediol (BDF). The ability to decompose PHB/BDF blends under the action of chemical, industrial lipases and fungal cultures at various concentrations of the plasticizer was studied. Particular advances have been made in assessing biodegradation in the case of fungal degradation using image processing, as well as monitoring of O2 consumption and CO2 production. Although no biodegradation was found with industrial lipases, it was found with chemical and fungal degradations, and this effect increased with increasing of plasticizer. Without any doubt, the article will be of interest for the readers of Polymers.

Thank you for your comments.

I think that the article can be improved taking into account the following comments and suggestions:

- The strength of the study is that the authors used three types of blends: specimen, film and powder. It is necessary to clearly indicate in the captions to the figures in which cases which types of samples were used (Figs. 2,4,5,6,8,9).

Indeed, for each figure we added precisions regarding the form of the blend that was used.

- Lines 150, 159; Equations 1, 2. Why not use the term Degree of Degradation instead of Degradation.

We modified the name in the two equations.

- What can be the accuracy of determining the values presented in Fig.2. Is the difference between samples with 30 and 40% plasticizer statistically significant?

Unfortunately, the experiments were carried out without replications. For that reason, we are unable to determine significant differences between the conditions.

- It is better to reduce the size of the drawings. It is necessary to correct expressions like g.L-1 to  g∙L-1.

The size of the figures was reduced and all the units were modified accordingly.
